Small-scale drivers on plant and ant diversity in a grassland habitat through a multifaceted approach

Mugnai Michele 1 michele.mugnai@unifi.it
Frasconi Wendt Clara 1
Balzani Paride 1 2
Ferretti Giulio 1
Dal Cin Matteo 1
http://orcid.org/0000-0001-5473-4649 Masoni Alberto 1
Frizzi Filippo 1
Santini Giacomo 1
http://orcid.org/0000-0003-3422-5999 Viciani Daniele 1
http://orcid.org/0000-0001-6451-4025 Foggi Bruno 1
Lazzaro Lorenzo 1
1 Department of Biology, University of Florence , Firenze , Italy
2 Faculty of Fisheries and Protection of Waters, South Bohemian Research Center of Aquaculture and Biodiversity of Hydrocenoses, University of South Bohemia , Vodňany , Czech Republic
Fenu Giuseppe
Electronic publication date: 2021 Dec 24
Publication date: 2021
Volume: 9
Electronic Location ID: e12517
Received 2021 Jul 23; Accepted 2021 Oct 27
Copyright: © 2021 Mugnai et al.
Copyright year: 2021
Copyright holder: Mugnai et al.
License: This is an open access article distributed under the terms of the Creative Commons Attribution License, which permits unrestricted use, distribution, reproduction and adaptation in any medium and for any purpose provided that it is properly attributed. For attribution, the original author(s), title, publication source (PeerJ) and either DOI or URL of the article must be cited.
License URL: https://creativecommons.org/licenses/by/4.0/

Keywords: Biodiversity, Functional traits, Multi-taxa, Priority habitat, Taxonomic diversity, Mediterranean, Community

Funding: The authors received no funding for this work.

==============================
Semi-natural grasslands are characterized by high biodiversity and require multifaceted approaches to monitor their biodiversity. Moreover, grasslands comprise a multitude of microhabitats, making the scale of investigation of fundamental importance. Despite their wide distribution, grasslands are highly threatened and are considered of high conservation priority by Directive no. 92/43/EEC. Here, we investigate the effects of small-scale ecological differences between two ecosites present within the EU habitat of Community Interest of semi-natural dry grasslands on calcareous substrates (6210 according to Dir. 92/43/EEC) occurring on a Mediterranean mountain. We measured taxonomic and functional diversity of plant and ant communities, evaluating the differences among the two ecosites, how these differences are influenced by the environment and whether vegetation affects composition of the ant community. Our results show that taxonomic and functional diversity of plant and ant communities are influenced by the environment. While vegetation has no effect on ant communities, we found plant and ant community composition differed across the two ecosites, filtering ant and plant species according to their functional traits, even at a small spatial scale. Our findings imply that small-scale monitoring is needed to effectively conserve priority habitats, especially for those that comprise multiple microhabitats.

Introduction

Semi-natural grasslands are habitats with very high species richness, representing biodiversity hotspots in temperate Europe (Wilson et al., 2012). The multitude of microhabitats sustaining a high biodiversity (Habel et al., 2013) justify the high conservation value of these habitats (Dengler et al., 2014). Alongside the richness of plant species, grassland environments are also important reservoirs of animal biodiversity (Zulka et al., 2014; Ambarlı et al., 2016), in particular of invertebrate groups (Van Swaay, 2002; Weiss, Zucchi & Hochkirch, 2013; Lacasella et al., 2015).

Despite their ecological importance, these habitats are undergoing a rapid decline due to abandonment, land use intensification and afforestation (Burrascano et al., 2013; Habel et al., 2013), which may lead to complex patterns of vegetation changes (Diekmann et al., 2019), including in some cases the simplification of their flora and fauna (Hilpold et al., 2018). This is particularly alarming, given that several threatened species belonging to different trophic groups are associated with these ecosystems (Dengler et al., 2014). For example, 59% of the 152 European bird species encountered in grasslands are threatened (Nagy, 2009). Therefore, grasslands are listed as high conservation priority in the Habitats Directive (Dir. 92/43/EEC).

Spatial scale has a critical role in structuring biodiversity (Irmler & Hoernes, 2003; Keil et al., 2017; Carey et al., 2017; Jarzyna & Jetz, 2018; Kral et al., 2018): changes at the small scale may not reflect changes in biodiversity at a larger scale (Crawley & Harral, 2001; Hooper et al., 2005; Chase & Knight, 2013), and different abiotic and biotic filters, acting at different spatial scales, may have a determinant role structuring community composition (Menegotto, Dambros & Netto, 2019). For example, in grassland habitats and at large spatial scale, factors such as management types and intensity determine differences in vegetation structure (Auestad, Rydgren & Økland, 2008). On the other hand, at a small scale, plant species composition resulted to be more affected by edaphic factors, e.g., soil moisture, depth and nutrients content (Auestad, Rydgren & Økland, 2008; Turtureanu et al., 2014). Furthermore, biodiversity metrics may respond differently to the spatial scale of analysis. For example, while species richness is generally considered one of the main measures of biodiversity, this metric is scale dependent (Belmaker & Jetz, 2010) and may not always depict changes in biodiversity at different spatial scales (Hewitt et al., 2010; Chase et al., 2019). Thus, rather than species richness alone, trait-based metrics may be used in a complementary way, as they may better inform on biodiversity changes across different spatial scales (e.g., Wong, Guénard & Lewis, 2019).

Traits have a universal approach, as they do not rely on species taxonomic identity, they inform on ecosystem multifunctionality and services and they more easily and rapidly respond to environmental changes (Violle et al., 2014; Gagic et al., 2015). Moreover, grassland communities’ functional composition has been shown to respond also to spatial scale of investigation. For example, De Bello et al. (2013) found that from the finest (within 50 × 50 cm plots) to the broadest scale (area of 22 km2), plant species traits ranged respectively from different values (divergence), to similar values (convergence), and finally to no significant differences. Additionally, plant traits are related to different environmental factors, which may influence plant community composition. For example plant height (H) is related to competition for light (Díaz et al., 2016), seed mass (SM) to competition among seedlings and chances of establishment (Westoby, 1998; Díaz et al., 2016), specific leaf area (SLA) to resource exploitation (Wright et al., 2004), leaf dry-matter content (LDMC) to environmental productivity and disturbance (Hodgson et al., 2011; Pérez-Harguindeguy et al., 2013) and leaf area (LA) hints around the implications on the regulation of leaf temperature and water use efficiency during photosynthesis (Díaz et al., 2016). Recently, Pierce et al. (2017) showed that the measurement of only three leaf traits (LA, SLA and LDMC) can effectively represent interspecific variation in plant size and conservative vs. acquisitive resource economics and that these can be used to reasonably deduce the position of individuals in the CSR framework according to Grime’s Competitive Stress-tolerant Ruderal (CSR) theory (Grime, 1977; Grime & Pierce, 2012).

To assess the ecological role of animal community within grassland habitats, many different groups of invertebrates have been proposed and used (e.g., De Deyn et al., 2003; Barber et al., 2017; Horváth et al., 2019). Among these, ants are shown to be particularly suitable. Ants are important ecosystem engineers, because they modify their surrounding environment and upper and lower trophic levels (Wills & Landis, 2018; De Almeida et al., 2020) and they are used as ecological indicators, because they “respond (or anticipate) to environmental change and represent other taxa” (Ellison, 2012; Frasconi Wendt et al., 2021b). Besides competition, climatic and environmental variables influence ant communities and predict changes in ant biodiversity at regional and local scale (Sanders et al., 2007; Dunn et al., 2009; Dröse et al., 2019; Frasconi Wendt et al., 2020). Accordingly, ant functional traits provide important ecological information, for example: Weber’s length (WL) is indicative of ant body size and is related to ant metabolism and thermal tolerance (Verble-Pearson, Gifford & Yanoviak, 2015); head length (HL) may be indicative for ant diet preferences, with granivorous species being characterized by longer heads (Yates et al., 2014); relative leg length (RLL), which corresponds to the leg length divided by head length, is related to body size, according to the size-grain hypothesis (Kaspari & Weiser, 1999), and on the way ants perceive their environment (Gibb & Parr, 2013); diet trait stands for food preferences and is influenced by resource availability (Arnan, Cerdá & Retana, 2014); trait behaviour refers to the dominant and subordinate behaviour of ants and is related to the thermal tolerance (Cerdá, Retana & Cros, 1998). For this reason, the inclusion of multiple taxonomic groups allows to better disentangle the processes shaping the high biodiversity of grassland habitats (Turtureanu et al., 2014).

Indeed, ants’ and plants’ diversity indices were shown to be good ecological indicators for the whole grassland community (Peters et al., 2016). These two communities are strictly connected and as habitat heterogeneity increases, so does resource availability, nesting possibilities and microclimatic conditions that may in turn support a higher species richness and functional diversity (e.g., Nooten et al., 2019). Furthermore, recently Caddy-Retalic et al. (2019) reported a decoupled response of plant and ant community composition under climate change, thus special attention should be given to these two taxa.

To assess how small-scale differences drive changes in communities’ structure, both in terms of species and functional traits, we investigated a species-rich semi-natural grassland (Habitat 6210 Semi-natural dry grasslands and scrubland facies on calcareous substrates (Festuco-Brometalia); Habitats Directive). In particular, we aimed to verify: whether small-scale ecological differences within the same habitat affect plants and ants. Particularly we expected that small-scale differentiation in two different ecosites comprised in Habitat 6210 individuates two ecological contexts, which may affect plant and ant species and trait composition;

the relative relationship among these key groups and the environment, following the framework developed by Frenette-Dussault, Shipley & Hingrat (2013). In particular, we expected that the different environments affect plant community composition and functions, and that the environment influences indirectly ant communities through changes in plant assemblage. This included assessing which ant traits are influenced by the environment too.

Materials and Methods

Study site

The study was conducted in the protected area of “Monti della Calvana” (43.915659N, 11.165627E), a Special Area of Conservation (SAC) under Habitat Directive (Natura 2000 code: IT5150001). The study area is a calcareous mountain range located at around 40 km from Florence (Italy) with a maximum elevation of 916 m a.s.l. Annual mean temperature is 6.2 °C and annual precipitation 1,023 mm (Karger et al., 2017). The sampling was conducted in the summital area of the landform, within grasslands classified as Habitat 6210 according to the HaSCITu (Habitat in the Sites of Conservation Interest in Tuscany) program (http://www.regione.toscana.it/-/la-carta-degli-habitat-nei-siti-natura-2000-toscani) and affected by low grazing intensity. A grassland patch of about 1.5 km2 was outlined within the habitat and, through satellite photo interpretation, it has been divided into two (of 0.75 km2 area each) contiguous but definable ecosites, characterized by different substrates (Fig. 1). One ecosite was characterized by inclination >15° and shallow soils with outcropping rocks (Rocky hereafter) while the other one for inclination <15° and deep soils (Deep hereafter). Despite the whole area was covered by the same grassland habitat, it resulted as easily divisible through photo interpretation into the two ecosites thanks to their topographical distinction (i.e., Deep consisted in the summital part of area while Rocky consisted in the south facing slope of the landform). The distinction into two ecosites was considered as a differentiation at a small scale of the grassland habitat present in the area.

Figure 1 Map of the study area.

Map of the study area in Monti della Calvana (Italy) and its division into two ecosites.

Sampling

Vegetation survey and sampling for plant traits were carried out in May-July 2019. For each ecosite we randomly placed 10 (20 in total) 1 × 1 m2 quadrat plots in which we surveyed plants and measured environmental variables. Following EDGG protocol (European Dry Grassland Group, see Dengler et al., 2016), plant species cover was visually estimated with a precision of 0.1%. As environmental variables, we measured microrelief (perpendicular distance in cm between lowest and highest ground points), vegetation cover (%), litter cover (%), soil depth (average from 5 replicates in cm) and vegetation height (average from 5 replicates in cm). We selected a set of leaf traits (LA, SLA and LDMC) together with plant height (H), given their responses to environmental factors at a small spatial scale. In addition, we included seed mass (SM) for its ecological relevance and for its link with myrmecochory and predation by ants (Boulay et al., 2007; Thomson et al., 2010; Pearson et al., 2014). For each plot we measured functional traits of plant species whose coverage summed up to the 80% of the total vegetation coverage (50 plant species in total), considering that they functionally represent the whole flora occurring in the plot (see Pérez-Harguindeguy et al., 2013). For each of these plant species, we measured in the field the height H of five individuals that were also collected; leaves were immediately immersed in cool deionized water and processed for measurement within 24 h after collection. We subsequently measured the following traits: (a) Leaf Fresh Weight (LFW); (b) LA, measured after digitizing the leaf outline (1,200 dpi) using ImageJ v. 1.51 software (Schneider, Rasband & Eliceiri, 2012) and (c) Leaf Dry Weight (LDW), after 72 h at 70 °C in an oven. Leaves’ weight was measured with an analytical balance, accurate to 0.01 mg. For each leaf, we calculated the SLA according to the formula SLA = LA/LDW and the LDMC according to the formula LDMC = LDW/LFW. LA, SLA and LDMC were measured for five individuals of each species and 3–5 leaves for each individual. Values of SM were retrieved from LEDA (Kleyer et al., 2008) and BROT (Tavşanoğlu & Pausas, 2018) databases. Leaf functional trait values of two species (Erodium cicutarium and Carduus nutans) were not obtained from field sampling, therefore they were retrieved from TRY (Kattge et al., 2020) and BROT (Tavşanoğlu & Pausas, 2018) databases.

Ant sampling was carried out in July 2019. We placed one pitfall trap at the corner of the same quadrat plots used for vegetation surveys, for a total of 40 pitfall traps per ecosite. Pitfall traps consisted of 50 ml Falcon® tube, with a diameter of 30 mm, filled with an 80% diluted ethanol and 2% glycerol solution and buried in the ground so that the rim of the trap was at the same level as the soil surface. The traps were re-collected after seven days and brought to the laboratory, where we identified ants to the species level (Czechowski et al., 2012; Lebas et al., 2016).

To assess the functional composition of ant communities, we used the following functional traits: Weber’s length (WL), head length (HL), relative leg length (RLL), diet (generalist, seed or sugar based) and behaviour (dominant or subordinate). For the continuous traits (WL, HL and RLL) we measured and averaged them on 15 individuals per species. When species accounted for less than 15 individuals, we measured all available specimens. As for polymorphic species, such as Messor capitatus, M. ibericus and Pheidole pallidula, we measured 10 individuals per caste and then the values were averaged per species. We retrieved the categorical (diet) and binary (behaviour) traits from the available literature (Arnan, Cerdá & Retana, 2014; Parr et al., 2017; Frasconi Wendt et al., 2020).

Data analysis

To assess whether the two different ecosites individuated through photointerpretation actually reflect different ecological conditions, we used two types of analyses. First, we performed an analysis of variance (ANOVA) after checking the assumptions (using Shapiro-Wilk normality test, Bartlett test for of homogeneity of variances and Durbin-Watson test for autocorrelated errors) for each environmental variable to check for differences between ecosites. Litter cover has been log-transformed before analyses. The only variable that did not fulfill all the assumptions was vegetation cover, for which we used the Welch One-way ANOVA (not assuming equal variances). Second, we calculated Grime’s CSR ecological strategies for each plot based on plant leaf traits. We used the StrateFy tool (Pierce et al., 2017) to obtain CSR values for each plant species and subsequently we calculated the Community-Weighted Mean (CWM, see below) of such values for each plot. Using CSR values as coordinates, plots were displayed in a ternary diagram (Pierce et al., 2017). In order to test whether the two contexts differ in terms of ecological strategies, each of the three CSR values were tested using ANOVA, after assessing the assumption of ANOVA. Only R coordinates did not fulfill all the assumptions, hence the Welch One-way ANOVA (not assuming equal variances) were performed. Moreover, R coordinates were log-transformed before analyses.

We used ant incidence data, corresponding to the number of pitfall traps in each quadrat that contained a given species, as ant abundance data may be biased by the proximity of the pitfall traps to an ant nest (Gotelli et al., 2011).

Plant and ant functional structure were assessed using the CWM, which measures the mean trait value for the whole community weighted by the abundance of the species carrying the trait (Garnier et al., 2007).

Differences in plant and ant community compositions between the two habitat ecosites were evaluated using non-metric multidimensional scaling (nMDS) based on the Bray-Curtis dissimilarity index (McCune, Grace & Urban, 2002) and their significance was assessed throughout a permutational multivariate analysis of variance (PERMANOVA) with 9,999 permutations. Subsequently, CWMs of plant and ant traits were fitted onto species ordinations using function envfit (Oksanen et al., 2020) to evaluate the direction and magnitude of traits variation within the two communities.

To measure the relationship between environmental variables, plant, and ant communities, we performed simple and partial Mantel’s tests, as proposed by Frenette-Dussault, Shipley & Hingrat (2013). In simple Mantel tests, correlations are performed between two distance matrices, whereas the partial Mantel test assesses the relationships between two distance matrices, while controlling for a third matrix. We used Bray-Curtis dissimilarity for the plant and ant species distance matrices, and Euclidean distance for environmental (Rocky and Deep ecosites) and traits (CWMs) distance matrices. Environmental matrix was built using distances between ecosites, while for the trait matrix we used CWMs of plant and ant traits.

Finally, to evaluate the relationship between environmental variables, ant species and ant functional traits, we performed a fourth-corner analysis (Brown et al., 2014) followed by the least absolute shrinkage and selection operator (LASSO) to obtain the most parsimonious model (Wang et al., 2020). This analysis, which provides a model for species abundance (in our case species incidence) as a function of environmental factors, functional traits and their interaction, quantifies the strength and the significance of the environment-trait interactions and identifies which interactions drive species abundances. The resulting matrix (fourth-corner) shows the interaction coefficients between the environmental variables and the functional traits (Brown et al., 2014).

nMDS and Mantel tests were conducted within the R package vegan (Oksanen et al., 2020) and the fourth-corner analysis within package mvabund (Wang et al., 2020). All analyses were performed using R 3.5.2 (R Core Team, 2017).

Results

The two ecosites resulted as characterized by similar species richness, for both plant and ant communities, with a slightly higher number of species in Rocky ecosite: 94 plant and 16 ant species in Deep ecosite, and 98 plant and 19 ant species in Rocky ecosite. In Information S1 are provided the sample-size-based rarefaction curves for both ecosites and plant and ant communities.

The environmental variables used to describe the two ecosites showed a significant difference between Rocky and Deep ecosites. In particular, herb coverage and soil depth were higher in Deep ecosite, while microrelief was higher in Rocky ecosite (Fig. 2 and Table 1). Moreover, CSR values between ecosites were statistically different (Table 1), with plots from Deep ecosite more characterized by R strategy and Rocky plots by C and S strategies (Figs. 2 and 3).

Figure 2 Variation of environmental variables and plant ecological strategies between the two ecosites.

Variation of environmental variables (scale in cm for soil depth, microrelief and vegetation height, and % for herb, and litter coverages) and plant ecological strategies between the two ecosites. Litter cover and R strategy are log-transformed.

Table 1 ANOVA statistics on differences between the two ecosites.

ANOVA statistics on differences between the two ecosites. An asterisk (*) denotes results of Welch ANOVA.

	Df	F value	P value	
Environmental variables	Microrelief	1	4.78	0.044	
Herb coverage (%)*	1	35.06	<0.001	
Litter coverage (%)	1	3.02	0.101	
Soil depth	1	24.83	<0.001	
Vegetation height	1	0.51	0.485	
Ecological strategies	C strategy	1	6.35	0.023	
S strategy	1	10.6	0.005	
R strategy*	1	30.16	<0.001	

Figure 3 Ecological strategies of communities using CSR values.

Ecological strategies displayed in a ternary diagram using CSR values.

Both plant and ant communities showed significant differentiation in species composition between Rocky and Deep ecosites, as the points representing them were mapped at the opposite extremes of the ordination space (Fig. 4, Table 2). The nMDSs showed a high goodness of fit already with two dimensions both for plant (stress 0.15, non-metric R2 = 0.976) and ant communities (stress 0.12, non-metric R2 = 0.987). Moreover, traits fitted onto ordinations showed some patterns in relation to the two ecosites, and most of them were statistically significant (Table 3). In particular, regarding plant traits, Rocky ecosite results characterized by higher LA and H, while for ant traits we found higher values of WL, HL, RLL, subordinate behaviour and generalist diet in Deep ecosite.

Figure 4 Community plant and ant ordinations resulted by nMDS.

Community ordinations resulted by nMDS, according to plant (A) and ant (B) species and with respective functional traits fitted onto ordinations. Green points correspond to plots from Deep ecosite, while blue points represent those from Rocky ecosite. Abbreviation: LA; leaf area, SLA; specific leaf area, SM; seed mass, H; plant height, LDMC; leaf dry matter content, WL; Weber’s length, HL; head length, RLL; relative leg length, Sugar Diet; sugar-based diet, Seed Diet; seed-based diet, Subordinate; subordinate behaviour, Dominant; dominant behaviour.

Table 2 Plant and ant community composition.

Results of PERMANOVAs tested on plant and ant community compositions (nMDS).

Community	Sum Sq	F value	R2	P value	
Plant species	1.760	9.651	0.376	<0.001	
Ant species	1.533	9.800	0.38	<0.001	

Table 3 Functional traits in plant and ant communities.

Results of tests evaluating magnitude and directions of functional traits onto plant and ant species compositions.

Functional trait	NMDS1	NMDS2	R2	P value	
Plant	LA	0.445	0.895	0.240	0.130	
LDMC	0.528	−0.849	0.514	0.005	
SLA	−0.952	−0.305	0.099	0.464	
H	0.469	−0.883	0.425	0.014	
SM	10.579	−0.815	0.246	0.122	
Ant	WL	−0.713	0.701	0.792	<0.001	
HL	−0.730	0.684	0.803	<0.001	
RLL	−0.252	0.968	0.619	0.001	
Generalist diet	--0.729	−0.684	0.376	0.026	
Seed diet	0.841	0.541	0.819	<0.001	
Sugar diet	−1.000	0.001	0.059	0.636	
Subordinate behaviour	−0.763	0.647	0.586	0.002	
Dominant behaviour	0.763	−0.647	0.586	0.002	

Simple and partial Mantel tests indicated two similar scenarios for taxonomic and functional community composition of both groups (Table 4). Correlations indicated that the environment (ecosites Rocky and Deep) but not vegetation composition had a direct effect on ant taxonomic and functional communities. Despite the taxonomic and functional approaches having similar outcomes, the first one shows a stronger correlation for environment-vegetation than for environment-ant, while the latter one reveals a stronger association for environment-ant rather than for environment-vegetation.

Table 4 Relationships between environment and plant and ant communities.

Relationships among environment, vegetation and ant communities considering species and functional traits. In both cases, correlation values above the diagonal correspond to simple Mantel tests, while correlation values below the diagonal resulted from partial Mantel tests. In brackets are reported P values and significant correlations are in bold. Following Frenette-Dussault, Shipley & Hingrat (2013), we considered as significance threshold α = 0.10.

Taxonomic	
	Environment	Vegetation	Ant	
Environment		0.75 (<0.001)	0.561 (<0.001)	
Vegetation	0.67 (<0.001)		0.461 (<0.001)	
Ant	0.366 (<0.001)	0.074 (0.227)		
Functional	
	Environment	Vegetation	Ant	
Environment		0.104 (0.074)	0.576 (<0.001)	
Vegetation	0.133 (0.048)		−0.009 (0.510)	
Ant	0.581 (<0.001)	−0.084 (0.772)		

The fourth-corner analysis results showed some significant relationships between ant functional traits and environment variables (Fig. 5). For example, body size traits were correlated with variables that describe environmental heterogeneity, with WL positively correlating with herb coverage. We also found a strong negative correlation between granivorous diets and soil depth.

Figure 5 Relationships between environmental variables, ant species and ant traits assessed through fourth-corner analysis.

On y axis are reported ant functional traits, while on x axis are reported environmental variables. Colors correspond to interaction coefficients (positive and negative) as reported in the bar on the right.

Discussion

According to our results, within the same grassland habitat and at a small-scale, we could identify two ecosites in which ecological differences led to different plant and ant communities. The discrimination of the two ecosites, which was initially based on photointerpretation, geomorphology (i.e., slope) and on a physionomical basis (i.e., vegetation coverage), was confirmed by ecologically different and relevant variables and by different CSR profiles at a small-scale. Indeed, CSR strategies are known to respond to ecological constraints at various spatial resolutions, at global (Pierce et al., 2017), medium (Rosenfield, Müller & Overbeck, 2019) and local scale (Negreiros et al., 2014; but see Rosado & de Mattos, 2017). The absence of other ecological factors, e.g., soil pH, soil moisture and thermal conditions, may represent a limitation to our study. Such ecological factors are known to shape plant and ant communities, as they directly influence nutrient and water availability, movement and ability to find food resources (e.g., Wiescher, Pearce-Duvet & Feener, 2012; Dengler et al., 2014; Arnan et al., 2015). However, the environmental variables used in this study are of primary importance in driving structure of grassland communities and widely used in similar research (e.g., Bernard-Verdier et al., 2012; Turtureanu et al., 2014).

The two ecosites were characterized by distinct plant and ant communities. Plant species composition is well known to respond to ecological differences at different scales (Auestad, Rydgren & Økland, 2008; De Bello et al., 2013; Turtureanu et al., 2014). In our case, Deep ecosite is characterized by nearly exclusive species (Lolium perenne, Cerastium glomeratum, Salvia pratensis, Phleum pratense, Centaurea solstitialis) with high ruderality (Pignatti, Guarino & La Rosa, 2017). Plant species composition at Rocky ecosite is more variable and with a few species typical of unproductive rocky grasslands (e.g., Trifolium stellatum and Crepis neglecta). These results are consistent with the ecological characterization accomplished through CSR strategies. With regards to ants, we found two different ant communities in the two ecosites, suggesting that ant assemblage reflects different habitats and conditions even at small-scale, as found by other studies (Frasconi Wendt et al., 2021a; Frizzi et al., 2021). We sampled mainly generalist species (e.g., Formica cunicularia and Myrmica spp.) in the Deep ecosite, whereas the Rocky ecosite was characterized by dominant species (such as Tapinoma erraticum and T. madeirense). These results support some recent studies from karstic environments in Hungary showing that both ant and plant species and functional composition (in terms of environmental requirements) respond to fine-scale microhabitat differences, namely between the inner parts of natural depressions (dolines) and the surrounding plateau areas, providing different microclimatic conditions and a high environmental heterogeneity (Bátori et al., 2019, 2020).

The environmental differentiation between the two ecosites was related to both plant and ant taxonomic and functional composition. The influence of the environment on plant composition can be explained by the ecological differences between the two ecosites. Soil depth and microrelief are two important drivers of plant community assembly both taxonomically and functionally (Bernard-Verdier et al., 2012; Turtureanu et al., 2014). Moreover, the environmental differences between the two ecosites may allow to infer on the distinct ecological responses by the plant communities. Shallower soils with a prevalence of abiotic fraction (i.e., outcropping rocks, stones and gravel; namely Rocky ecosite), selected for more stress-tolerant plant communities, whose species deal with less resources availability (mainly water) and for this reason result in higher LDMC values. Similarly, steeper slopes are characterized by lower water availability and host communities in which environmental filtering is selecting more stress-tolerant sets of traits, such as higher LDMC (Bricca et al., 2021). This effect could be increased also by the south-facing aspect of Rocky ecosite, according to the relationship between topographic exposure and vegetation extensively reported in literature (e.g., Li et al., 2011; Valencia et al., 2015). On the contrary, the Deep ecosite is characterized by flatter and deeper soils, where soil resources are more abundant, and vegetation thrives and forms dense cover. Here, plant species showed higher SLA and communities were characterized by a higher ruderality, probably linked to higher levels of disturbance. Indeed, despite both ecosites are subjected to a low intensity of grazing, Deep ecosite could be more affected by disturbance as a result of the low inclination of slope, which not only mitigates the north-facing aspect but also offer more viable paths for hikers and livestock.

While the direct effect of the environment on vegetation was expected and extensively demonstrated, the direct effect of the environment on ant community composition is debated. For example, Frenette-Dussault, Shipley & Hingrat (2013) findings highlight the presence of a direct effect of vegetation and only an indirect effect (through vegetation structure) of the environment on ant species and trait composition (Frenette-Dussault, Shipley & Hingrat, 2013). On the contrary, other studies support the trait-environment constraints and the role of the environment acting as the primary filter on ant composition (Wiescher, Pearce-Duvet & Feener, 2012; Boet, Arnan & Retana, 2020). Indeed, the strong environment-ant association and the role of environmental variables influencing ant taxonomic and functional diversity at different scales has been acknowledged in past studies and in different ecosystems too (Wiescher, Pearce-Duvet & Feener, 2012; Frenette-Dussault, Shipley & Hingrat, 2013; Arnan, Cerdá & Retana, 2014, Arnan, Cerdá & Retana, 2015; Nooten et al., 2019). Microrelief, slope inclination, vegetation height and cover, and soil composition are important variables describing the ecological conditions occurring in grassland habitats where sampled ants live and forage, and they are good predictors of ant community composition and diversity along environmental gradients in different ecosystems (Debuse, King & House, 2007; Pacheco & Vasconcelos, 2012; Arnan, Cerdá & Retana, 2014; Mauda et al., 2018). For example, Deák et al. (2020) found that grassland habitats characterized by steeper slopes and more topographically heterogeneous host more diverse arthropod communities, with ant being one of the main groups. Therefore, the distinct ant communities may be explained by the different environmental compositions of the two ecosites (Bátori et al., 2020), meaning that at small-spatial scale, environmental parameters, which may describe the microhabitat and microclimatic conditions, are stronger than vegetation at driving ant community composition. Nevertheless, it must be considered that the evaluation of other vegetation features may better explain the actual variation of ant communities.

The fourth-corner analysis revealed that ant body size was not related to the microrelief and instead increased with herb cover, contradicting past studies (Nooten et al., 2019) and the size-grain hypothesis (Kaspari & Weiser, 1999): having large body size may represent an impediment for the movements and foraging of ants in a denser environment, such as the one characterized by a higher herb cover, whereas in a more open and planar environment, body size increases (Gibb & Parr, 2013). Two possible explanations may be given for these contrasting results. First, our spatial scale of analysis, although at a small-scale, may not fully reveal the habitat heterogeneity and soil rugosity as perceived by ants. Nooten et al. (2019) suggested that contrasting outcomes of the environment-ant relationship may arise because of the different ways habitat complexity is measured in distinct studies. Second, other unmeasured variables, such as climatic factors, rather than herb cover per se may explain changes in ant body size (Wiescher, Pearce-Duvet & Feener, 2012; Arnan, Cerdá & Retana, 2014). The other strong relationship found in the fourth-corner analysis, concerned the negative association between seed-based diet and soil depth. The dominance of the seed-based diet in Rocky ecosite, which is characterized by shallow soils, may be driven by the higher occurrence of harvester ant species belonging to the genus Messor compared to Deep ecosite. In fact, species of the Messor genus were collected exclusively in a few sampling points in the Deep ecosite. Other soil characteristics, such as texture, clay content and chemistry may be better related to ant diversity (Bestelmeyer & Wiens, 2001; Debuse, King & House, 2007). From a conservation perspective, our results value the importance of protecting the local rich-plant and -ant communities within each ecosite of this semi-natural grassland. This may be achieved through the maintenance of habitat heterogeneity, namely different microclimatic and microhabitat conditions, which seem to play an essential role in the conservation of plants and ants in similar grassland ecosystems too (Bátori et al., 2019). However, these fine-scale microhabitat differences and the biodiversity associated with this SAC site may be threatened because of the loss or intensification of low-intensity anthropogenic practices. This is particularly alarming, given that the site is under the Habitat Directive (Natura 2000 code: IT5150001) and a SAC, thus, efforts should be deployed to protect this grassland ecosystem even at a small-spatial scale.

Conclusions

We demonstrated that small-scale ecological differentiation into ecosites of a grassland habitat affects key groups, i.e., plants and ants, in terms of species composition and functional traits. Furthermore, we found that the environment directly influenced taxonomic and functional composition of both groups, while vegetation seemed to play a minor role in shaping the assembly of ant communities, at least at a small-scale. These results highlighted the importance of using complementary biodiversity approaches (i.e., taxonomic and functional). Moreover, plant and ant taxa revealed to be suitable key groups for multi taxa studies. Lastly, our results showed that considering only the habitat type in biodiversity studies can lead to coarse approximations, thus emphasizing the importance of small-scale monitoring of key biodiversity groups for the effective assessment of environmental changes on biodiversity.

Supplemental Information

Supplemental Information 1 Sample‐size‐based rarefaction sampling curves.

Click here for additional data file.

Supplemental Information 2 Raw data: raw measurements used for all the analyses.

1) plant community data, with percentage coverage on plot for each species; 2) ant community data, with incidence in plot for each species; 3) plant traits, with mean value of considered traits for the species that accounted for the 80% of cover in each plot; 4) ant traits, with mean value of considered traits for each species occurring in the plots; 5) environmental variables measured at plot level.

Click here for additional data file.

The authors want to thank Fabrizio Rigato for ant species confirmation.

Additional Information and Declarations

Competing Interests

Author Contributions

Field Study Permissions

Data Availability

The authors declare that they have no competing interests.

Michele Mugnai conceived and designed the experiments, performed the experiments, analyzed the data, prepared figures and/or tables, authored or reviewed drafts of the paper, and approved the final draft.

Clara Frasconi Wendt conceived and designed the experiments, performed the experiments, analyzed the data, authored or reviewed drafts of the paper, and approved the final draft.

Paride Balzani conceived and designed the experiments, performed the experiments, analyzed the data, authored or reviewed drafts of the paper, and approved the final draft.

Giulio Ferretti performed the experiments, authored or reviewed drafts of the paper, and approved the final draft.

Matteo Dal Cin performed the experiments, authored or reviewed drafts of the paper, and approved the final draft.

Alberto Masoni performed the experiments, authored or reviewed drafts of the paper, and approved the final draft.

Filippo Frizzi performed the experiments, authored or reviewed drafts of the paper, and approved the final draft.

Giacomo Santini conceived and designed the experiments, analyzed the data, authored or reviewed drafts of the paper, and approved the final draft.

Daniele Viciani conceived and designed the experiments, authored or reviewed drafts of the paper, and approved the final draft.

Bruno Foggi conceived and designed the experiments, authored or reviewed drafts of the paper, and approved the final draft.

Lorenzo Lazzaro conceived and designed the experiments, performed the experiments, analyzed the data, authored or reviewed drafts of the paper, and approved the final draft.

The following information was supplied relating to field study approvals (i.e., approving body and any reference numbers):

The area in which we performed the surveys is an ANPIL (Protected Natural Area of Local Interest) defined by Tuscany region. Hence, the area is fully accessible to the public and there is no need of any authorization to visit the area, as well as to perform studies on its territories or to collect sample materials (except for the current legislation on fauna and particular flora species). Our study did not involve the collection of any species of flora and fauna under particular regime of protection.

The following information was supplied regarding data availability:

The raw measurements used for all the analyses are available in the Supplemental File.

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
