# Peer review of "Small-scale drivers on plant and ant diversity in a grassland habitat through a multifaceted approach"

_PeerJ, doi:10.7717/peerj.12517_

## Round 0.1 · original submission · Major Revisions

Dear authors
I received two very different reviews on your paper; the main problems highlighted by reviewer 1 concern the limited data sets, the lack of measurement (or modeling) of environmental gradients and the lack of separation effects of management from land survey. I agree with him/her that these limitations negatively affect the study; however, also considering the highly positive evaluation of reviewer 2, I ask you to try to fill all main gaps highlighted by reviewer 1 and to resubmit a new version of your paper, revised following all the suggestions of the two reviewers.

Please consider my decision as an opportunity to strongly improve the paper; I will carefully consider the new version of the manuscript before making a new decision.

Reviewer 1 ·

Basic reporting

The article documented differences in vegetation and ants between two closely placed sites, which differ considerably regarding environmental conditions (flat area with deep soil layer vs. southern slope with shallow soil). It was a good idea to examine two species groups: both ants and vascular plants. The examined habitat (semi-natural grasslands) are among the most important for biodiversity maintenance, and simultaneously, strongly endangered habitats in agricultural landscapes of the European Union. The article is well written, the discussion is supported by results. The quality of figures and tables is appropriate

Experimental design

The topic is within the journal scope. The research question were well defined, the data were collected using up-to-date methodological approach, and the stastistical methods of analysis were correct.

Validity of the findings

The data are restricted to a single location only, and the significant effect of microrelief on vegetation structure and ants communities is already known. Because of examination of only one location, it is unclear to what extent the observed pattern is idiosyncratic. Moreover, there was no attempt to assess the magnitude of differences in ecological conditions (e.g. thermal conditions, moisture, soil parameters) between the two studied sites, except for soil deep. It caused that some of the conclusions, underlying effect of local differentiation within a habitat, are less useful. It is hard to assess how the contrast between microsites was strong, and to what extent the obtained results can be compared with studies from other sites/regions. Complementing the examination by measured or modeled environmental variables and including it in the analysis will increase the value of the research and allow to write a more specific discussion. Additionally, the observed pattern emerges not only from differences in landrelief but also from management intensity (Discussion: lines 307-310). It would be better to find study sites under similar management type and intensities.

Additional comments

Additional comments
• Line 43 – it is not so easy; see Diekmann et al (2019)
Diekmann, M., Andres, C., Becker, T., Bennie, J., Blüml, V., Bullock, J. M., ... & Wesche, K. (2019). Patterns of long‐term vegetation change vary between different types of semi‐natural grasslands in Western and Central Europe. Journal of Vegetation Science, 30(2), 187-202.
• Line 53. The management type is management, not an environmental factor. I suggest not confound it, especially that in management type its intensity, not only type, is important.
• Line 83. How did You define ‘the ecological status of ecosystems’ here? This term is used rather in hydroecology. It sounds nice, but what it means exactly?

Materials & Methods
• Line 137: ‘steeper slope’ –describe it using inclination in degrees, please.
• Line 198. Does the data fulfill the ANOVA assumptions (e.g distribution, variance homogeneity)
• Line 200. Provide a reference for the StrateFy tool

Results.
• The map of local landrelief and sampling plots location should be provided
• A brief description of the taxonomic diversity of studied communities will be helpful for the readers (e.g alfa and total diversity). Also, it will be good to add in the supplementary materials the rarefaction curves, for eco-sites and taxonomic groups.

Discussion
General comments. In the discussion, the effect of slope exposition is omitted. As is described in Materials & Methods, the Rocky area was places on southern slopes. It could be assumed that it considerably changed thermal conditions of compared sites, as well as influence the water conditions (higher evaporation on southern slopes). This issue should be discussed. Moreover, there is correlation between landrelief and management, thus its hart to divide the effect of the environment from management influence. This issue should be discussed.

• Lines 269-270. I guess that the differentiation between the eco-sites was also easy to indication using the digital terrain model and its derivatives
• Line 304. What did You mean as ‘abiotic fraction’ of soil?
• Lines 311 – 313. The second part of the sentence repeats information from sentence in lines 295-298
• Lines 315 – 319. It seems that the sentence is contradictory to massage from lines 295-298. Can You explain it in more detail?

• Lines 319-324 The plant height is a trait of vegetation (plants) and can not be shown as a variable describing the soil-plant interface. What do you mean by ‘habitat cover’? Moreover, I think that it will be better to not consider microrelief with soil. It could be considered as two different groups of factors

Conclusion.
The two first sentences, provides generally the same information. Thus, one from them is redundant and should be deleted.

Supplementary materials
The supplementary materials (excel file) should be prepared more carefully. There are no full names of ant species, only abbreviations (contrary to plants), lack of units in plant and ants traits, as well as environmental variables.
If the plant species cover from the ‘plant_species’ sheet is summed up, the value differs from the herb cover showed in the ‘environmental_var’ sheet. If you estimated the plant species cover with 0.1% accuracy, these values should be more or less similar. Is it OK?
Photographs documenting the general appearance of vegetation/habitats in bot studied eco-sites will be helpful.

Reviewer 2 ·

Basic reporting

In this work, author measured taxonomic and functional diversity of plant and ant communities at two semi natural dry grassland ecotypes, evaluating the differences among the two ecosites, how these differences are influenced by the environment and whether vegetation affects composition of the ant.

Experimental design

156-157. How did you estimate species abundance? Visually, using a grid? I would change a bit this sentence since to estimate vegetation cover at the 0.1% is not an easy job.
160. On how many individuals per species did you measure SLA? Also specify here.

Validity of the findings

Despite the limited sampling strategy and the small area, the work is valid and well written and well designed, while statistical analyses are appropriate.

Additional comments

Minor comments
164. “traits of plants” or better “traits of plant species”. In any case from here and from the results it’s not clear the number of species on which you performed the measures. Please, specify.
Fig. 1/Results. Finally, I would add two plots describing the difference in alpha (richness) and beta diversity between the two ecosites. In addition, at the main difference in species composition should be shortly described at the beginning of the Results. You partly describe such aspect in the discussion but no mention in the result

---

## Round 0.2 · accepted · Accept

Although the manuscript still shows some gaps (e.g. limited number of field observations, and lack of environmental data), which could not be filled at this time, I suggest accepting the paper for publication in consideration of the relevance of the study presented.

Reviewer 1 ·

Basic reporting

-

Experimental design

-

Validity of the findings

-

Additional comments

I would like to thank the Authors for their careful reply to the comments and changes made in the manuscript. However, I can not agree with the statement that ‘measuring other environmental variables in the field a posteriori is not possible, since the communities were sampled in 2019 and ecological conditions might be changed’. It is hard to believe that soil pH, insolation, and water conditions will change within 2 years period. A simple, yet meaningful quantification of environmental variables (e.g. stable water conditions, insolation) can be obtained from digital elevation model, without any additional field works. Nice examples You can find in articles of Moeslund and al. (2013) and Kopecký et al. (2021)

A Moeslund, J. E., Arge, L., Bøcher, P. K., Dalgaard, T., & Svenning, J. C. (2013). Topography as a driver of local terrestrial vascular plant diversity patterns. Nordic Journal of Botany, 31(2), 129-144.
Kopecký, M., Macek, M., & Wild, J. (2021). Topographic Wetness Index calculation guidelines based on measured soil moisture and plant species composition. Science of The Total Environment, 757, 143785.

Reviewer 2 ·

Basic reporting

no comments

Experimental design

No comments

Validity of the findings

no comments

Additional comments

I found this version of manuscript really improved according to the reviewers' comments